# Silver Nanoparticles: Synthesis, Structure, Properties and Applications

**DOI:** 10.3390/nano14171425

**Published:** 2024-08-31

**Authors:** Rimsha Abbas, Jingjing Luo, Xue Qi, Adeela Naz, Imtiaz Ahmad Khan, Haipeng Liu, Suzhu Yu, Jun Wei

**Affiliations:** 1Shenzhen Key Laboratory of Flexible Printed Electronics Technology, Harbin Institute of Technology (Shenzhen), Shenzhen 518055, China; rimsha.abbas31@stu.hit.edu.cn (R.A.); 21b355005@stu.hit.edu.cn (J.L.); qixue@hit.edu.cn (X.Q.); 23bf55002@stu.hit.edu.cn (A.N.); 23bf55003@stu.hit.edu.cn (I.A.K.);; 2School of Materials Science and Engineering, Harbin Institute of Technology (Shenzhen), Shenzhen 518055, China; 3State Key Laboratory of Advanced Welding and Joining, Harbin Institute of Technology (Shenzhen), Shenzhen 518055, China

**Keywords:** silver nanoparticles, synthesis, Ag NP’s structure, properties, NP applications

## Abstract

Silver nanoparticles (Ag NPs) have accumulated significant interest due to their exceptional physicochemical properties and remarkable applications in biomedicine, electronics, and catalysis sensing. This comprehensive review provides an in-depth study of synthetic approaches such as biological synthesis, chemical synthesis, and physical synthesis with a detailed overview of their sub-methodologies, highlighting advantages and disadvantages. Additionally, structural properties affected by synthesis methods are discussed in detail by examining the dimensions and surface morphology. The review explores the distinctive properties of Ag NPs, including optical, electrical, catalytic, and antimicrobial properties, which render them beneficial for a range of applications. Furthermore, this review describes the diverse applications in several fields, such as medicine, environmental science, electronics, and optoelectronics. However, with numerous applications, several kinds of issues still exist. Future attempts need to address difficulties regarding synthetic techniques, environmental friendliness, and affordability. In order to ensure the secure utilization of Ag NPs, it is necessary to establish sustainability in synthetic techniques and eco-friendly production methods. This review aims to give a comprehensive overview of the synthesis, structural analysis, properties, and multifaceted applications of Ag NPs.

## 1. Introduction

Nanotechnology is the investigation of metals in the size range of 1 to 100 nm. The world of nanoscience deviates from the larger world we are accustomed to in our daily lives, centered on numerous aspects of nanotechnology. When the length scale of materials decreases, surface-area effects become more important and quantum effects emerge, leading to modifications in the properties of materials. The quantity of atoms appropriate for an object’s surface increases due to the compact nature of metals [1]. Nanoparticles (NPs) demonstrate incredible diversity in chemical, size, charge, shape, and surface area [2].

At these sizes, the properties of NPs are distinct from those of the bulk. Particle size distribution and morphology reveal unique and enhanced characteristics at the nanoscale. NPs possess characteristics common to both distinct phases of particles and solutes. When compared to larger particles or atoms, NPs have a surface-to-volume ratio that is 35–45% higher [3,4]. Various nanocarriers have been reported, including liposomes, peptide-based NPs, dendrimers, polymer-based NPs, quantum dots, carbon nanotubes, and, most importantly, metal nanoparticles (MNPs). NPs provide innovative applications in drug delivery, biosensors, microarrays, microfluidics, and tissue micro-engineering for the specialized treatment of ailments [5]. The accomplishments of new NP applications and the corresponding enhancements in research and development have a direct impact on the economy and society. It has been investigated that nanotechnology, in the form of NPs, exerts numerous impacts on various industries, including information technology, life sciences, and electronics [6].

Almost all areas of the natural sciences today, including biology, optics, catalysis, and sensing, utilize metallic nanoparticles [7]. MNPs are ideal catalysts for a variety of catalytic processes because of their specific characteristics. This area grabs the attention of researchers due to the integration of innovative behaviors, inventive catalysts, and recent advances [8]. MNPs, especially Ag NPs, have exceptional properties that enhance their attractiveness to society. The extraordinary physical, chemical, and biological properties of Ag NPs have been the main focus of research. The primary reason for their ubiquity is the mismatch between the bulk structure and the form, composition, crystallinity, and structure of Ag NPs [9]. Due to their size-dependent optical and catalytic capabilities, Ag NPs have vast applications in biological fields. Since Ag NPs do not penetrate human skin, they have been utilized in industrial usage as safe preservatives in a variety of cosmetics [10]. The statistics showing the requirements and expenditure in Ag NP-related research indicate the attention these molecules have garnered. With an expected annual output of over 500 tons of NPs to meet the demands of several industries, the market for Ag NPs has been constantly expanding over the past 15 years [11]. It has been reported that the size, distribution, morphological shape, and surface characteristics of Ag NPs have a strong impact on their catalytic, physical, and optical properties. These characteristics can be altered using a variety of synthetic techniques, including capping and stabilizing agents. The morphology of Ag NPs depends on the particular application. For example, Ag NPs made for drug delivery are typically larger than 100 nm to allow for sufficient drug supply. Ag NPs have a variety of surface characteristics that enable them to take on numerous forms, such as rod, round, triangular, octahedral, and polyhedral shapes [12].

The stabilizing/capping agents have a strong impact on the shape and size of Ag NPs. Different capping and stabilizing agents have been reported for the surface functionalization of Ag NPs. The production of Ag NPs generally involves several types of physical and chemical methods, which fall into one of two categories: ‘top-down’ or ‘bottom-up’. Green synthesis techniques have recently been devised and used to generate Ag NPs without the use of hazardous chemicals [13]. The synthesis of NPs using physical and photochemical processes typically requires expensive equipment, as well as extremely high temperatures and vacuums. The majority of the current techniques for producing Ag NPs are chemical in nature. The chemical method is the most widely used due to its convenience and minimal need for expensive equipment. This approach can successfully generate uniformly sized Ag NPs. By the chemical approach, Ag NPs are synthesized in solutions under mild and simple conditions. Colloidally dispersed Ag NPs are produced by chemical reduction in water or organic solvents [14]. Silver nanowires (Ag NWs) and Ag NPs provide enhancements in electrical conductivity and the ability to function as photonic devices, inks, pastes, and fillers. Their optical qualities offer great benefits in biosensing [15].

However, for the formation of uniform nanostructures, more attention is currently being paid to creating nanomaterials with regulated morphologies and nanoscale dimensions to achieve the desired results. More developments are expected, with nanomaterials being integrated into next-generation electronics to meet high energy demands in the future, and actively participating in biosensors and nanomedicine to combat diseases. Due to a lack of knowledge, one of the primary obstacles in contemporary nanotechnology is finding substitutes for the use of restricted and perilous resources in the formulation of nanomaterials [16]. Producing MNPs with control over their size and structure is a serious challenge. Many approaches require high temperatures and harsh chemicals, which affect the yield rate for large-scale production of MNPs. Additionally, achieving uniform size and shape control for MNPs is challenging because these parameters significantly impact their properties and applications. Environmental toxicity of MNPs has been discussed in various studies, with some MNPs, including Ag NPs, having the potential to harm aquatic life. Further studies on the effects of MNPs on the environment are essential, as is the development of green synthesis and processing techniques. A better understanding of the challenges facing contemporary society and the rapid development of nanotechnology can help alleviate future problems [17].

This review discusses several approaches for the creation, structure, properties, and applications of Ag NPs. Ag NPs offer vast applications in the fields of science and technology worldwide. The primary objective of developing Ag NPs techniques is to take advantage of their unique properties, such as optical, electrical, high surface-to-volume ratio, and antibacterial capabilities. Several approaches, including chemical, physical, and biological methods, are used to synthesize Ag NPs. Each technique has specific advantages in terms of cost-effectiveness, environmental friendliness, and scalability. Ag NPs find numerous applications in multiple fields such as electronics, biological sensing, environmental remediation, and catalysis. They exhibit extraordinary antimicrobial activities that are beneficial in drug delivery and diagnosis. Their high surface area and unique morphology make them efficient catalysts for a variety of catalytic processes. Additionally, their optical properties offer benefits in sensing, and their antibacterial activity finds use in disinfecting and purifying water. Overall, the creation of Ag NPs contributes to the advancement of technology and the innovation of new techniques.

## 2. Synthesis Techniques, Structure and Properties of Ag NPs 

The large-scale and high-yield production of Ag NPs with flexible particle shapes and sizes has been the main focus of recent investigations. However, achieving NPs with consistent sizes and shapes remains a global challenge [18]. The remarkable characteristics of nanomaterials depend on their dimensions, configuration, interactions with stabilizers and surrounding media, and methods of fabrication. Therefore, achieving the desired properties of NPs requires controlled synthesis of nanocrystals. The size, shape, and chemical environment of NPs determine their optical, magnetic, electrical, and catalytic capabilities [19]. There have been several ways to synthesize Ag NPs since the advent of nanotechnology. A wide range of current techniques can be categorized into two fundamental approaches for synthesizing Ag NPs: top-down and bottom-up [20] (Figure 1).

According to the top-down approach, several physical forces, such as mechanical processes like crushing, grinding, and milling; electrical methods like electrical arc discharge or laser ablation; and thermal techniques like vapor condensation, are used to generate metal NPs from bulk materials. Physically produced NPs are highly pure and often exhibit a homogeneous distribution of particle sizes. However, this method does not involve potentially harmful chemicals or stabilizing agents to prevent agglomeration, but it requires sophisticated equipment and significant external energy. In contrast, the bottom-up approach involves assembling molecular constituents into complex aggregates through growth and nucleation. Chemical and biological synthesis are common bottom-up techniques for producing NPs from precursor salts. Chemical synthesis can enhance efficiency by combining light, electricity, microwaves, or sound waves. NPs produced chemically can be rapidly generated and applied in various configurations. However, their use in medical applications may be limited due to the potential risks associated with the synthesis process involving hazardous substances. Biological synthesis, which has become increasingly significant over the past few decades due to its ecological advantages, utilizes chemicals derived from microbes such as alcohols, flavonoids, alkaloids, quinines, terpenoids, and phenolic compounds, as well as cellulose, enzymes, and exopolysaccharides [21,22,23].

### 2.1. Biological Methods

Traditional techniques for producing NPs are expensive, hazardous, and environmentally unfriendly. To circumvent these issues, researchers have explored green routes—naturally occurring sources and their products that can be used to synthesize NPs. Biological synthesis can be classified into several methods, including the use of microorganisms such as fungi, yeasts (*eukaryotes*), bacteria, and actinomycetes (*prokaryotes*). Another approach involves utilizing plants and plant extracts, as well as templates such as viruses, membranes, and diatoms. Natural biologically active substances can be found in plenty in shells and peels of food waste. Synthesis of Ag NPs using food wastes is beneficial as compared to chemical synthesis [24]. The green synthesis of Ag NPs utilizes agricultural waste and by-products from the food industry as eco-friendly reducing and stabilizing agents. Examples include banana peels, orange peels, and potato skins, which are rich in natural compounds such as polyphenols and flavonoids. These compounds reduce silver ions to form NPs and help stabilize them, reducing the need for harmful chemicals. This method not only offers a sustainable approach to nanoparticle synthesis but also adds value to agricultural and industrial waste, supporting waste management and environmental protection [25].

The following sections provide descriptions of biological synthesis techniques using fungi, bacteria, and plant extracts as shown in Figure 2.

Biological methods involve two processes: biosorption and bio-reduction [26]. The biological synthesis of Ag NPs requires a reducing biological agent and a silver metal ion solution as the main ingredients. Typically, there is no need to add external capping and stabilizing agents because the reducing agents or other components already present in the cells act as these substances’ stabilizing and capping agents.

#### 2.1.1. Plant-Mediated Synthesis

The process of producing NPs involves the following steps: the plant of interest is harvested from its natural habitat, thoroughly washed with tap water two or three times to remove necrotic plants and *epiphytes*, and then rinsed with deionized water to eliminate any remaining debris. After cleaning, the plant parts are dried in the shade for ten to fifteen days before being ground into powder. To prepare the plant extract, approximately 10 g of the dried powder is soaked in 100 mL of deionized distilled water and heated using the hot percolation method. The resulting infusion is filtered to remove any insoluble material. A few milliliters of the plant extract are added to a 10^−3^ M AgNO_3_ solution, causing pure Ag^+^ ions to be reduced to Ag^0^, which can be monitored by periodically measuring the solution’s UV-visible spectra [27].

In the synthesis process, plant extracts can be used as capping agents. For example, Kumar, D.A. et al. capped Ag NPs using *Alternanthera dentata* plant extract during the synthesis process [28]. Kumar, S. et al. also demonstrated the mixture of plant leaf extracts with silver nitrate (AgNO_3_) solution using *Parthenium hysterophorus* leaf extract and *Premna herbacea.* Spherical shaped Ag NPs with sizes 10–30 nm show potential anti-bacterial activities against *Shigella dysentrieae* and *Escherichia coli* (*E.coli*), two Gram-negative bacteria that cause dysentery in humans [29]. Manjamadha and MuthuKumar used a weed plant for ultrasound-assisted synthesis of Ag NPs. The use of ultrasound increases the reaction rate in a short time [30]. The aqueous extract of *Peganum harmala* leaves was used by Taghrid S. Alomar et al. to synthesize Ag NPs, for the preparation of the aqueous extract, leaves of the *Peganum harmala* plant were collected and soaked in hot water. The resulting mixture was then mixed with an AgNO_3_ solution. A color change from colorless to brown indicated the formation of Ag NPs after stirring the reaction mixture at a specific temperature. This eco-friendly technique stabilizes silver ions, enabling Ag NPs to potentially find applications in biomedical and pharmaceutical fields. The study concluded that this synthesis technique enhanced the photoluminescence properties of Ag NPs [31]. The most commonly used plant species for synthesizing Ag NPs are *Tephrosia purpurea*, *Sesbania grandiflora*, and *Morinda citrifolia*. These plants possess great phytochemical properties, which make them highly beneficial for treating diseases [32]. Hui Xu et al. reported the synthesis of Ag NPs from food waste using grape seed extract as a reducing and stabilizing agent within 10 min. These Ag NPs, with a zeta potential of −28.4 mV, show high stability. The synthesized Ag NPs, with sizes ranging from 25 to 35 nm, demonstrate potential applications against eight types of Gram-negative and Gram-positive bacteria [33]. Sharma et al. synthesized Ag NPs using various vegetable peels, which were boiled for 10 min. The vegetable peel extract powder was prepared by crushing, filtering, and treating the filtrate with cold ethanol. After adding Ag NO_3_ to the powdered vegetable extract, the mixture was incubated, leading to the synthesis of Ag NPs [34]. Georgii Vasyliev et al. used apricot and black currant pomace as waste materials to create aqueous extracts for the green synthesis of Ag NPs. The zeta potential of the obtained colloidal solutions ranged from −33.41 to −24.23 mV, indicating moderate stability of the synthesized NPs. These NPs effectively demonstrated a bactericidal effect against *E. coli* [35].

#### 2.1.2. Microbial Synthesis

S.V. Otari et al. reported the synthesis process of Ag NPs using the actinobacteria *Rhodococcus* sp., which is a green biosynthesis process. *Rhodococcus* sp. is used to reduce aqueous silver nitrate. This synthesis process leads to the formation of Ag NPs with a uniform size of 10 nm, providing various applications in fields such as biological labeling, antibacterial activity, and catalysis [36]. Lihong Liu et al. demonstrated a revolutionary method for synthesizing Ag NPs using microorganism culture broth without necessitating any specific living microbe. This study highlights the significance of pH levels, light, and broth composition for the production of pure Ag NPs. It investigated the formation of Ag NPs without living microbes under suitable light and pH conditions, which has great significance in nanomaterial synthesis [37]. Mohd Yusof et al. used *Lactobacillus plantarum* TA4 to synthesize Ag NPs while tolerating Ag^+^. They found that the cell biomass of *L. plantarum* TA4 has the ability to tolerate Ag+ at a concentration of 2 mM. The presence of maximum UV–Vis absorption centered at 429 nm and the observation of color changes confirmed the formation of Ag NPs [38].

#### 2.1.3. Bio-Polymer-Mediate

Swarup Roy et al. proposed a green method for synthesizing Ag NPs using melanin (Mel) as a reducing and capping agent, and antimicrobial nanocomposite films were prepared by combining them with carrageenan (Carr). The stability of Ag NPs was indicated by a hydrodynamic radius of 59.51 nm and a zeta potential of 31.03 mV. It was investigated that thermal stability increased with a low concentration of Ag NPs in the film, and the maximum decomposition temperature was 257 °C. Mechanical parameters such as elongation at break (E), elastic modulus (EM), and tensile strength (TS) were affected by the addition of Ag NPs [39]. Ag NPs were synthesized using a chitosan/chitin-based technique. These Ag NPs served as nuclei for the production of silver nanowires (Ag NWs) on a drop-cast chitosan/chitin thin film. Irregular twisted Ag NWs were produced, which proved to be more beneficial in various chemical detection systems [40]. In this study, *Adenia hondala* was used to synthesize Ag NPs, which were then coated with chitosan and loaded with the medication tamoxifen. The drug releasing efficacy improved with the decrease in pH from 7.4 to 4.0 [41].

#### 2.1.4. Enzyme Assisted Synthesis

Biosynthesis has gained more interest due to its economically viable and sustainable techniques. A recent study reported the synthesis of Ag NPs with a size of 5–10 nm using *Rhizoctonia solani* fungi, which demonstrated strong antibacterial properties again *S. aureus* [42]. Ag NPs were synthesized using an enzyme-induced reduction method, a simple wet-chemical process that does not require complicated patterning or vacuum deposition [43]. Enzyme-assisted hydrolysis was employed to extract non-extractable ferulic acid from oats by-products, specifically rye bran. Ag NPs were not generated using pure synthetic trans-FA under unbiased conditions until sodium hydroxide (NaOH) was added, resulting in alkaline formation. However, this study did not explore the biocompatible and cytotoxic properties of Ag NPs generated from rye bran [44]. In another approach, Ag NPs were synthesized using enzyme-assisted extracts obtained from plants and fungi. The study provides a comparison between the synthesis methods and antibacterial properties of Ag NPs formed by the *pseudocereal F. esculentum* and *lichen C. islandica* (using raw and enzyme-assisted extracts) [45].

### 2.2. Chemical Methods

The chemical method is the most widely used approach for synthesizing Ag NPs due to its high effectiveness and low cost. There are several approaches to synthesizing Ag NPs using the chemical method, such as electrochemical methods, chemical vapor deposition, chemical reduction, and reverse micelle techniques. Among these, chemical reduction is the most commonly employed approach [46]. The chemical synthesis process typically requires three main components: a reducing agent, capping/stabilizing agents, and a precursor (Figure 3). The solvent serves as the fourth component. The most commonly used chemical reactions for synthesizing Ag NPs include borohydride reduction [47,48], the citrate method [49,50], the polyol process [51,52], and the Tollens reaction [53]. Initially, the citrate method was applied for synthesizing Ag NPs and proved to be very effective in exploring the behavior of Ag NPs [45]. However, borohydride reduction offers explicit control over the shape and size of Ag NPs due to its excellent reducing capacity compared to the citrate method [54,55].

Sodium borohydride (NaBH_4_) is commonly used as a reducing agent in borohydride reduction, and precise control over its use allows for the production of various sizes and shapes of Ag NPs such as spheres, triangles, and rods using the same set of chemicals [54]. In this review, we will explain various chemical synthesis approaches for Ag NPs.

#### 2.2.1. Sol–Gel Method

The sol–gel method is an efficient chemical approach for producing sophisticated materials in a variety of research fields. When combined with techniques such as phase separation, hybridization, and templating induction, this method provides greater control over size and shape, which is highly innovative for various applications [56]. Siloxane surfactant treated with glucose was used as a stabilizing and reducing agent for the formation of Ag NPs through redox reactions. This study examines the synthesis of spherical Ag NPs, which have an average diameter of 6.5 nm when synthesized without glucose and 14 nm when synthesized with glucose [57]. Ag NPs with a clean surface were synthesized using the sol–gel method at room temperature. In this process, sodium acetate (CH_3_COONa) was used to prevent aggregation of Ag NPs, and hydrazine was used as a reducing agent. The produced NPs had an average size of 11 nm, and their crystallinity and crystal plane orientation were confirmed using X-ray diffraction (XRD) analysis, which matched the standard pattern for nano silver. Scanning Electron Microscopy (SEM) results indicated that the particles were uniformly sized, homogeneous, and exhibited clean, well-defined granular shapes, free of contamination within the nanoscale range [58]. The NPs were synthesized using a hydrolytic sol–gel approach within silica matrices. The study investigated the potential applications of Ag NPs in plasmonic solar cells [59].

#### 2.2.2. Hydrothermal Method

Ag NPs were first synthesized by the hydrothermal method using bacterial cellulose (BC) as both a stabilizing and reducing agent. Narrow distribution of Ag NPs from 17.1 ± 5.9 nm [60]. Hydrothermal green synthesis of Ag NPs was performed using Pelargonium/Geranium leaf extract without the use of toxic chemicals. Response surface methodology (RSM) was used to generate experimental models for the λ max coloration of the synthesized Ag NPs solution, with the amount of 1 mM AgNO_3_ solution and *Pelargonium/Geranium* leaf extract concentration (PLEC) as dependent variables [61]. Ag NWs were synthesized by hydrothermal methods, and this study investigated the antibacterial activities of Ag NWs [62]. A one-pot hydrothermal technique was used to synthesize Ag NPs and reduced graphene oxide (RGO) nanocomposites. It was reported that Ag NPs-RGO nanocomposites provide potential antioxidant properties [63].

#### 2.2.3. Chemical Vapor Deposition (CVD)

For the first time, similar bacterial strains were distinguished based on their lipidomic patterns, showing strong potential for investigating antibiotic resistance using Ag NPs substrates generated using CVD [64]. This study reported the single-step manufacturing of a heterostructure formed by concentrated Ag NPs (size 2–10 nm) and chemical vapor deposited graphene as a surface-enhanced Raman scattering (SERS) substrate. The CVD graphene surface was coated with Ag NPs in a single step, where pure (99.98%) Ag foil was dissolved in diluted nitric acid, reducing the need for additional toxic chemicals and providing an eco-friendly technique for device construction. It was investigated that the generated hybrid nanostructure of Ag NPs could serve as a SERS substrate for numerous applications such as photovoltaic and electromagnetic devices, gas sensors, and electronics [65].

#### 2.2.4. Electrochemical Synthesis

This study reported the synthesis of Ag NPs using an electrochemical approach, with Poly (N-vinyl-2 pyrrolidone) (PVP) and sodium lauryl sulfate (Na-LS) employed as stabilizing and co-stabilizing agents. The novelty lies in the purportedly “sacrificial anode” process [66,67]. Stable Ag NPs were synthesized by an electrochemical method [68]. By changing the current polarity within sodium polyacrylate (Na PA) solutions, Ag NPs were produced using electrolysis with silver electrodes. It was reported that the polydispersity of Ag NPs increases with a decrease in the observed proportion of growth and nucleation, while the average size of Ag NPs clusters decreases due to an increase in the observed nucleation rate [69]. Green tea leaves and a bulk silver strip were used to synthesize biogenic colloidal Ag NPs via a green electrochemical method. It was investigated that biogenic Ag NPs have potential applications in electrochemical sensing [70]. This technique is not suitable for the large-scale production of Ag NPs [71].

#### 2.2.5. Microemulsion Method

The dispersity and size of the created Ag NPs strongly depend on the soluble capacity of the reducing reagents [72]. Several pieces of literature reported the use of chemicals to synthesize Ag NPs by microemulsion technique [73]. This study reported the production of stable and homo-disperse spherical Ag NPs with a size of about 3–10 nm using reverse microemulsion polymerization and the reverse microemulsion technique. Reverse microemulsion polymerization is a quick and effortless technique and can also be used to generate other kinds of MNPs [74]. Methodology to synthesize Ag NPs using water explained [75]. This work provided the use of triton X-100 (TX-100) and cetyltrimethylammonium bromide (CTAB) in W/O microemulsion to generate Ag NPs. In this method, NaBH_4_ is used as a reductant and AgNO_3_ is the antecedent. Noted that Triton X-100 provides more stable Ag NPs as compared to Cetyltrimethylammonium Bromide (CTAB) [76]. Green synthesis of Ag NPs with sizes ranging from 25 to 150 nm has been reported; geranium leaf aqueous extract is used as a reductant in W/O microemulsion and nanoemulsion techniques. O/W nano-emulsions provides a variety of shapes of synthesized Ag NPs but more stability is provided by microemulsion [77]. Reactions of Ag NPs with curcumin in microemulsion were analyzed and concluded by providing strong potential in bioimaging and sensing [78]. Microemulsions (3a–f) based on benzyl alkyl imidazolium ionic liquids (BAIILs), a novel group used as stabilizers, and silver nitrates as a reductant are used to generate monodispersed Ag NPs. This is a new technique reported in which no agglomeration of NPs was founded [78]. Ag NPs created by dioctyl sodium sulfosuccinate (AOT) microemulsion were concluded to be faster-released therapeutic agents at cancer cells in contrast to the circulation of blood [79]. Investigated the storage ability of generated Ag NPs for six months by mixing silver acetate with oleyl amine reductant at 70 degrees Celsius [80,81].

#### 2.2.6. Chemical Reduction Method

The pH value of solution affects the size, shape, and color of Ag NPs in the chemical reduction method [82]. Trisodium citrate is used as a reducing agent to synthesize Ag NPs by the chemical reduction method. Various reducing agents can be utilized in the procedure to generate NPs of different sizes, each with distinct antibacterial properties [83]. Polyvinyl pyrrolidone (PVP)—Aloe Vera mixture used as reducing agents to synthesize Ag NPs for antibacterial activity [84]. The shape of NPs changes from quasi-spherical to polygonal if the rest of the Ag^+^ ions continuously start forming Ag^0^ and attach to the surface of existing Ag particles in the presence of a moderate reductant. The Ag NPs had an average size of 50 nm, with a size range of 35 to 80 nm. It was observed that increasing the concentration of trisodium citrate led to a decrease in nanoparticle size, whereas an increase in ascorbic acid concentration had the opposite effect, resulting in larger NPs [85]. The simplest, fastest, and most inexpensive chemical reduction method to synthesize Ag NPs was reported [86,87]. Less reactivity produces less agglomeration, although powerful reductants generate small NPs [80]. Cationic interchange reagents were utilized to reduce the concentration of Ag^+^ from natural Ag NPs while maintaining the quality of solution by extracting free silver ions from processed Ag NPs solution [88].

#### 2.2.7. Polyol Process

Xia and colleagues reported the polyol synthesis of Ag NPs, which is the simplest and most eco-friendly technique [89,90]. In this method, polyols are used as reductants for metal salts [91]. Constant synthesis of Ag NPs investigated using polyol process [92]. In polyol processes, solvents have the greatest control over the size of NPs [93,94,95]. Green synthesis approach for polyol method performed to generate Ag NWs [96]. Aminopropyl trimethoxy silane (APTMS) is used as a stabilizing agent in ethylene glycol media to synthesize hexagonal Ag NPs with a 50–100 nm size distribution [97]. Torras and Roig investigate the microwave assisted polyol technique to produce Ag NPs [98]. This technique enhanced 61% of the Ag NPs formation rate for every 1 mg clutch, and for 20 mg of each clutch, the formation rate will be more than 98% [98]. Microwave-assisted (MW-assisted) polyol technique performed for the formation of Ag NPs with higher mono dispersity and identical size [99]. Ag NPs generated under various chemical reactions in a short time by the cheapest polyol technique, providing potential applications for sensors [100].

#### 2.2.8. Photochemical Reduction

The photoreduction approach was used to produce Ag NPs in films of polymeric material [101]. A green approach was performed using tyrosine as a photo-reductant and water as a solvent, resulting in large hydrodynamic diameter and small particle dimensions [102]. A green photochemical reduction approach was used to produce Ag NPs in κ-Carrageenan under ultraviolet (UV) light interference [103]. This technique reports the synthesis of icosahedral Ag NPs using UV irradiation assisted by tartrate as a reducing agent, achieving a production rate of over 90% [104]. Monodispersed Ag NPs were produced using a ferritin photochemical approach [105]. *Pistacia khinjuk* leaf extract (*P. khinjuk*) was utilized as a reductant to ensure an eco-friendly photochemical reduction technique for the formation of Ag NPs. Transmission electron microscopy (TEM) examination revealed that the NPs had a face-centered cubic (FCC) structure with a homogeneous, uniform, oval-like, and spherical morphology and a size ranging from approximately 35 to 45 nm [106]. The simplest and low-cost technique was performed to generate iso-Ag NPs using furanocoumarin as a reductant [107]. Potato starch was used in the photochemical reduction method to synthesize Ag NPs, making the process cheapest and convenient, with starch acting as a stabilizer [108].

### 2.3. Physical Methods

This is a top-down approach for the production of Ag NPs, utilizing physical factors such as electromagnetic radiation, plasma, and heat [109,110,111,112]. These synthesis techniques include approaches like laser ablation, evaporation–condensation using a gas tube [113,114,115], and arc discharge, considered the fastest physical method for Ag NPs formation [116]. A plasmonic technique known as lithography provides high control over the size of the generated Ag NPs, but it is costly and laborious [117]. Physical methods used for large scale production are mostly in the form of ashes with a uniform size of Ag NPs [118]. We are going to discuss physical methods for synthesis of Ag NP as demonstrated in Figure 4.

#### 2.3.1. Sputtering

For the formation of nanocrystalline thin sheets and powders at high pressure, magnetron sputtering is considered a potential technique because it provides high control over the production rate of Ag NPs [119]. Oxidized Ag NPs were generated by involving two steps: thermal evaporation of Ag NPs and sputtering of oxidization clumps by plasma [110]. Photosensitive Ag NPs were generated by direct current (DC) sputtering in a titanium dioxide (TiO_2_) matrix [120]. Ag NPs/thin sheets synthesized by sputtering using discharge voltage upon canola and castor [121]. Investigated that the direct current magnetron sputtering produce Ag NPs with large control on size and shape. The average sizes of Ag NPs with a constant sputtering current and deposition period were 5.9 ± 1.8 nm, 5.4 ± 1.3 nm, and 3.8 ± 0.7 nm for target–substrate distances of 10, 15, and 20 cm, respectively. Additionally, the shape of the NPs evolved from discrete NPs to worm-like networks [122]. Sputtering metal onto the liquid discussed by magnetron sputtering of silver and titanium pentaerythritol ethoxylate (PEEL) or 1-butyl-3-methylimidazolium bis(trifluoro methane sulfonyl)imide (BMIMTFSI) ionic liquid (IL) resulted in the formation of Ag NPs [123]. Reported technique synthesized Ag NPs using DC sputtering by altering the timing of depositions [124]. This technique enhances the production ability of Ag NPs, providing high control over shape and size [125].

#### 2.3.2. Physical Vapor Deposition (PVD)

Physical vapor deposition (PVD) is composed of three steps: sublimation, transportation of material, and nucleation/formation of NPs [126]. Use of electron beam PVD technique reported for the production of Ag NPs (15–20 nm) in a salt-based mixture. Investigated that antibacterial properties are strongly dependent on annealing temperature [127].

#### 2.3.3. Laser Ablation

Pure N,N-dimethylformamide, acetonitrile, dimethyl sulfoxide, and tetrahydrofuran are used to synthesize Ag NPs without the use of any reductant or stabilizer [128]. In contrast to chemical methods, laser ablation provides high purity of generated Ag NPs. There is no need for any capping and stabilizing agents, and it is considered an eco-friendly approach. Due to this reason, it provided higher capabilities in microbial activities than the chemical method [129,130,131]. This approach provides strong situ coupling with biomolecules as compared to ex situ coupling for chemical methods [132]. Ag NPs were generated by a femtosecond laser ablation process with various agents like deionized water (DIW), double distilled water (DDW), dimethylformamide (DMF), and tetrahydrofuran (THF). Analyzed that formatted Ag NPs in DIW are more stable and have potential capability in microbial activities as compared to other agents [133]. Laser ablation in liquid is considered a more beneficial approach as compared to other approaches [131,134]. Jong-Wan et al. produced Ag NPs by laser ablation technique [135]. A coating is generated by the interactions between high energy lasers and isopropanol while synthesizing Ag NPs, which prevents the interactions of Ag NPs that ensure higher stability [136]. Neodymium-doped yttrium aluminum garnet (Nd: YAG) laser ablation process reported to produce Ag NPs provided antimicrobial properties. However, by changing laser settings, more applications could be expected [137]. This study reported the production of silver iodide NPs using a pulsed laser in water, providing potential bacterial capabilities [138]. This is an expensive approach and needed high utilization of energy for production of Ag NPs [139]. Also investigated were the properties of synthesized Ag NPs influenced by the types of lasers being used [140].

#### 2.3.4. Arc Discharge

This is one of the physical approaches for the production of Ag NPs. This process involves the elimination of arc in the mixture. However, it does not provide high control on shape [141]. Titanium electrodes are used to synthesize NPs using the arc discharge approach. AgNO_3_ reduces due to arc discharge by applying 15 A current while keeping electrodes in the AgNO_3_ mixture for six minutes [142].

#### 2.3.5. Spark Discharge

Spark discharge, with the involvement of silver electrodes, DC, and deionized water, ensures the production of stable colloidal NPs [143]. The benefit of this technique is that it provides stable suspension. This study investigated the toxicity of pure Ag NPs on the hydrophytic plant *Lemna minor* produced by spark ablation at a quantity less than 5 μgL^−1^ [144].

### 2.4. Photochemical Synthesis

The sources of light for this process include laser light, sunlight, and UV light [145]. In this technique, at the very beginning of this process, metal precursors reduce from n^+^ valence state (Mn^+^) to zero-valence state (M^0^) due to their photocatalytic properties [108,146]. The study reported the formation of Ag NPs using chitosan/clay in the presence of ultraviolet radiation. The modified chitosan film, which contains dodecyl and DEAE groups, displayed smaller and more uniform nanoparticle sizes, along with a mixture of exfoliated and integrated structures. This amphiphilic chitosan modification is effective in regulating the size and shape of the Ag NPs [147]. Multiple groups work on the formation of Ag NPs within ferritin [148,149]. Ferritin has been utilized as an electrode, which releases the electrons that reduce the metallic ions [150,151]. Ag NPs synthesized by ferritin using the photochemical reduction method reported strong antimicrobial activities [105]. The production of silver nano decahedrons (Ag NDs) was investigated in the presence of blue LED light [152]. A cost-effective technique adopted for the formation of Ag NPs using starch in the presence of ultraviolet radiation ensures the fast production of Ag NPs [108]. This technique synthesizes NPs in both bottom-up and top-down methods (Figure 5).

### 2.5. Pros and Cons of Different Synthetic Approaches of Ag NPs

Several biological, chemical, and physical approaches are used for the formation of Ag NPs [113]. A biological approach is considered eco-friendly due to the use of plant extract, fungi, and bacteria as reductants to generate Ag NPs [153]. The synthesis of Ag NPs from agri-food waste is considered highly effective due to its environmental benefits. Utilizing agri-food waste helps reduce pollution, as the procedure does not produce additional waste. This approach promotes sustainability by effectively reusing waste materials [25]. Due to the use of natural resources, biological methods are economical and convenient, and there is no use of costly and harsh chemicals [154]. However, various factors are necessary for consideration, such as catalyst order, attributes of organisms, optimum response, and genetic and inherited features of organisms for the stability of generated Ag NPs [155]. Biological methods provide a variety of shapes and properties of Ag NPs. They required minimum upfront expenditures, and after the process, no separation was needed. However, they are cytotoxic at the biomolecular level due to the presence of both Ag ions and Ag NP. It is difficult to produce large amounts of NPs using biological methods. The final products may contain impurities [14,20,156,157,158,159].

The chemical approach is the most prevalent, abundant, and most effective for the generation of Ag NPs [154]. Chemical reduction, electrochemical, and microemulsion are some of the most widely used chemical methods to synthesize Ag NPs [55,160]. The chemical approach ensures the thermal stability and regulation of the production rate of Ag NPs, and with the use of multiple stabilizers, it provides stability of the generated Ag NPs. On the other hand, the wet chemical method is considered a remarkable technique due to its precise control, simplicity, affordability, and wide range of Ag NPs [155]. Additionally, the chemical approach is a cost effective for large scale production, appropriate, and fast technique without complicated tools [154]. The generated NPs could be stored for a long duration with barely any loss in stability [161]. However, chemical approaches are contemplated as corrosive and energy intensive [162]. Furthermore, the synthesized NPs get stained with chemicals, and significant harmful effects are produced [163,164]. It is a time effective approach and provides a large production rate, but for the prevention of aggregations, toxic chemicals are utilized as a reductant and capping agent, such as sodium citrate and N, N-dimethylformamide. Due to the production of impurities, further purification is needed. This technique is sensitive to atmospheric parameters and provides a lower re-production rate [13,18,165]. The advantages and disadvantages of chemical methods are discussed in Table 1.

The physical approach is composed of various sub-methods to synthesize Ag NPs; the most powerful methods include arc-discharge, laser ablation, and PVD. Physical methods provide high size uniformity and purity of the produced Ag NPs. Physical methods are most effective for large scale production and generate Ag NPs in ashes. They avoid the use of toxic chemicals, which is considered an environmentally friendly approach, but aggregation is produced due to the lack of utilization of capping or reducing agents [21,23]. But the production rate of Ag NPs using conventional physical approaches is very low. Moreover, this synthetic approach required special tools [154,155,162,164]. It is a quick approach for the formulation of Ag NPs with uniformity in size. However, the main issue with this approach is to alter the physicochemical properties and surface level chemistry of NPs. Generated Ag NPs have a short lifetime with less thermal stability [4,118,155,156,165]. The advantages and disadvantages of physical methods are discussed in Table 2.

### 2.6. Structure and Properties of Ag NPs

As Ag NPs have a wide range of applications, it is necessary to study their properties, which are strongly dependent on the shape and size of NPs [184]. Infections of microorganisms, including molds, yeast, viruses, and bacteria, are most common in humans, due to which several antibacterial materials were discovered by researchers. MNPs are widely studied because they have large surface atoms and surface area and extraordinary properties such as optical, physicochemical, antimicrobial, magnetic, and electronic. Among MNPs, Ag NPs provide extremely high antibacterial properties [185]. Silver provides a large surface area for bacterial interactions, with NPs attached on the cell membrane and within the bacterium [185]. An electrostatic attraction is established between positively charged Ag ions of Ag NPs and negatively charged membranes of cells that leads to the attachment of Ag NPs with the cell wall or membranes of the subjected microorganisms [186]. It was reported that due to surface transformations, Ag NPs demonstrate mechanical antibacterial properties along with the inherent biological interference abilities [187]. In the medical field, multi-shaped Ag NPs were used, including rods, triangles, flowers, and spheres [188,189]. The most important properties of Ag NPs are called physicochemical properties, which include shape, surface area, surface charge, etc. Smaller particles have a large surface area [190,191]. For Ag NPs, surface energy has a linear relation with surface area, which ensures the enhancement of biological properties [192]. Ag NPs synthesized by a wet chemical method for the treatment of Gram-negative bacteria. Investigated that antibacterial properties are strongly shape- and size-dependent. Small sized spherical Ag NPs show high antibacterial capabilities, while large sized spherical Ag NPs show less antibacterial capabilities as compared to triangular shaped Ag NPs [155]. Spherical facets (100) show less antibacterial properties in contrast with triangular facets (111) of Ag NPs [155,193]. The reason behind their remarkable anti-bacterial properties is that the bottom plane of anisotropic shaped Ag NPs having high atom-density with (111) facets leads to the highest anti-bacterial properties, provided the largest reactive area [194]. A recent study reported that triangular shaped nanoplates possess less antibacterial properties as compared to nanospheres towards *P. aeruginosa*, *E. coli*, and *S. aureus* [195]. Because Ag nanospheres provide greater contact with bacteria as compared to triangular nanoplates [196], Ag NPs, both quasi-spherical with size of 21 nm and spherical with size of 9 nm entirely provide anti-fungal properties [197]. Production of 5–20 nm Ag NPs by HEPES buffer reported, which provide antiviral properties. Further investigation revealed that Ag NPs of size 22 nm have strong wound healing capabilities [198]. Spherical-shaped Ag NPs with a size of 23.7 nm, synthesized using banana peel extract, exhibit potential antibacterial properties against common yeast and bacterial pathogens [199]. A green synthetic approach was adopted to synthesize spherical-shaped Ag NPs with a size of approximately 10.59 nm, using a non-edible part of the *Cynara scolymus* L. fruit. The Ag NPs exhibited antibacterial properties at small concentrations, ranging from 0.03 to 0.25 μg/mL. It was observed that at a concentration of 25 μg/mL, the Ag NPs produced about 50% inhibition on cancer cell lines [200]. The stability of Ag NPs is opposite to the glutathione reported by regulating the shapes with genetic sequencing [201].

Ag NPs have a wide range of applications on an industrial scale, including in sensors, due to their optical properties [202]. Ag NPs have extraordinary absorption and dispersion abilities due to their color, which changes according to the size and shape of NPs. These unique abilities of Ag NPs lead to the oscillations of conductive electrons on the surface of metal, referred to as surface plasmon resonance (SPR), initiated by light of a specific wavelength. Spherical Ag NPs have the distinct ability to change the SPR peak wavelength from 400 nm (violet) to 530 nm (green) by changing the size of the particle and the localized refracted index adjacent to the particle’s surface [171,185]. It was reported that spherical shaped Ag NPs with a size of 7 nm have SPR at a wavelength of 400 nm, though for 29 nm and 89 nm particles, SPR is at 425 nm and 490 nm. Concluded that SPR strongly related to the size of NPs [184]. Reliant upon the symmetry of NPs, irregular Ag NPs can display multiple SPRs [187]. Because of size dependency, type of material, and dielectric coefficient, local surface plasmon resonance (LSPR) is very useful in biological, chemical, and spectroscopic techniques [203]. Observed that Ag NPs with 400 nm SPR, spherical in shape, produced by glucose reduction, while 420 nm for NaOH reduction with the same morphology, resulted in strong applications in sensors and for improvements of solar cells as well [184].

Ag NPs generated in ceramic and glass have varying electrical conductivity due to size ranging from 4 to 12 nm. At 80–300 K, direct electrical resistivity of Ag NP film was investigated, resulting in a linear relation between temperature and surface resistivity from 120 to 300 K. Also investigated the linear relation between size of Ag NPs and Debye temperature [184]. The melting range for Ag NPs is from 4 to 50 nm, and thermal properties are investigated in sizes of 3–6 nm [184]. Investigated that stability of spherical shaped Ag NPs is smaller in contrast with hexagonal shaped Ag NPs [184]. TEM and SEM images for different shapes of Ag NPs are shown in Figure 6.

Properties of Ag NPs like optical, physical, chemical, and catalytic are strongly affected by the shape, size, and surface features, depending upon synthetic techniques and capping/stabilizing chemicals [171,206,207,208,209], as demonstrated in Table 3.

## 3. Applications of Ag NPs

### 3.1. Biomedical Applications

Due to physical properties such as size, shape, morphology, and surface area, magnetic properties, and electrical and optical properties, Ag NPs have wide applications in multiple fields [243], as shown in Figure 7. The main application of silver’s medical and preservation properties is to insulate the vessels against infections by bacteria and to keep water and other liquids reusable [244].

#### 3.1.1. Antiseptics

Ag NPs provide large antibacterial properties and are also very useful in decoding deoxyribonucleic acid (DNA) [245]. The antibacterial properties of Ag NPs are strongly shape- and size-dependent [194] Ag NPs exhibit antibacterial activity against *E. coli*, *Salmonella typhi*, *S. aureus*, and *Candida albicans* when synthesized by *Cryphonectria* sp. [189]. Ag NPs also proved antagonistic towards *Candida albicans* [246]. Ag NPs are extraordinary in defiance of human immunodeficiency virus and hepatitis B virus (HIV and HBV) [192,247]. Investigated that Ag NPs show less toxicity for the treatment of COVID-19 [248]. This is because Ag NPs attach to virus spikes in glycoproteins while preventing the attachment of viruses to cells [249]. It has been reported that Ag NPs generated by the medicinal plant *Azadirachta indica* provide excellent cardio protection in rats [250]. Karen M. Soto et al. reported the synthesis of Ag NPs using lyophilized extracts from grape and orange waste as reducing and stabilizing agents. The Ag NPs produced from both extracts exhibited minimal variation in growth inhibition of *L. monocytogenes*, with an inhibition diameter of 13.5 mm at 100 μg/mL. However, only the Ag NPs derived from the grape extract demonstrated dose-dependent antibacterial activity against *E. coli* O157, with a final OD of 0.42 at 100 μL [251].

#### 3.1.2. Drug Delivery Systems

Ag NPs could be used in sunscreen cosmetics and also provide beneficial effects in burn healing, dental appliances, and decoration of stainless steel objects [252]. The range of Ag NPs for the delivery of drugs is about 10–1000 nm, and due to their smaller size and larger surface area, they provide excellent benefits [253]. Ag NPs are more effective than other metal-based tiny materials in terms of extermination parameters and blue-shifting plasmon resonance peaks. Which renders them an excellent choice for applications: for example, surface-enhanced light chemistry of confined materials, like nitrobenzyl adjunct, and photo-controlled drug administration [254]. Nanobots provide enhancements in contrast with typical drug administration methods, like faster metabolism, extended plasma life, and endothelial-mediated targeted drug administration for tumor sites [255].

#### 3.1.3. Imaging and Diagnostics

In the last twenty years, NPs have been produced on a large scale for the enhancement of imaging techniques [256]. Ag NPs are used in the diagnosis of cancer cells and destroy them through photothermal treatment [257]. Existing studies reported the function of citrate-capped Ag NPs as a detector for the colorimetric assessments of creatine in human urine. This detection is most important for human health because, after detecting the kidney’s functions, it will be possible to apply a strong medical diagnosis [258]. Different methods were adopted for the determination of arginine using Ag NPs [259]. A recent study reported the use of an optical sensor for the measurement of nucleosides in human urine using Ag NPs [260]. Ag NPs particles provide unbeatable optical properties, due to which their use in diagnosis techniques has increased. In radiotherapy, Ag NPs increase the probability of destroying tumor cells [188]. Abhirami Santhosh et al. reported the antibacterial properties of Ag NPs synthesized using onion peels in a green approach. Furthermore, efforts have been made to create biosensors capable of detecting mercury, a hazardous metal, in the liquid phase [261].

### 3.2. Catalysis and Sensing

Ag NPs provide many catalytic applications. Ag NPs are widely used in a variety of sensors. They provide potential applications in printed electronics through the formation of inks.

#### 3.2.1. Catalytic Converters

The catalytic ability of Ag NPs was also determined by the reduction in dyes using silica spheres; in the absence of Ag NPs, no reduction in dyes was observed [262]. Ag NPs are used in textile materials by apparel and footwear sectors [263]. For chemical luminescence, Ag NPs behave as a strong catalyst as compared with gold and platinum emulsions [264]. Ag NPs provide photocatalytic properties opposite to those of color compounds such as naphthol orange (NO) and malachite green (MG) [156].

#### 3.2.2. Chemical Sensors

Colorimetric techniques composed of Ag NPs and Au NPs proved strongly precise and efficient in environmental investigations, especially when used in metal ions and biomolecule analysis [265]. It has been reported that Ag NPs provide applications in the sensing of lead (Pb) II ions, followed by interactions with dithizone [266]. Ag NPs are created by leaves of *Aconitum violaceum* used for the formation of colorimetric sensors for Pb (II). This tree biennial grows in Pakistan, Nepal, India, and the Himalayas [267]. For the detection of water impurities, Ag NPs were used [268]. Hydrogen peroxide (H_2_O_2_) is considered a harmful chemical compound, and Ag NPs are used as sensors for the detection of even small quantities of H_2_O_2_ and also for heavy metal pollutant detection [156].

#### 3.2.3. Environmental Remediation

Ag NPs have various applications in the environment, like in the purification of air, ground, and drinking water, and for the treatment of biological waste [269,270]. Ag NPs and combined materials lessen or eradicate the colorant, and this is very helpful to minimize environmental pollutants [271]. Ag NPs also have a lot of applications in farming as they influence the bacteria in the ground [272]. Ag NPs incorporated membranes made of nanocomposites have a strong ability to detoxify salts [273]. It is reported that in fifty percent of retail goods, twenty percent are made by nano-silver [274,275].

### 3.3. Electronics and Optoelectronics

Optoelectronic methodologies, like organic light-emitting diodes (OLEDs) and polymer light-emitting diodes (PLEDs), are widespread and integrated into our everyday lives. It has a variety of advantages, especially in communications through optical fiber, photonic converters, automation of devices, and in scientific research institutions [276]. We discussed several techniques for the formulation of Ag NPs with a variety of shapes and sizes, and both are the major parameters modulating the optical properties [244].

#### 3.3.1. Conductive Inks

The development of printable inorganic and organic materials such as insulative, conductive, and semi-conductive materials serves as the primary catalyst for flexible printed electronics (FPE). Due to their extraordinary oxidation resistance and high electric conductivity, Ag NPs are widely used in high-efficiency conductive inks as compared to other conductive materials. Ag NP’s based conductivity of printed electronics is affected by packability and the process of sintering. Packability depends upon the shape and size of NPs [277]. The electrical properties of the printed Ag NPs-based film are strongly dependent on the size and shape of the NPs. The size distribution of Ag NPs highly affects the sintering and electrical resistivity of printed designs [278,279,280,281,282]. Generally, spherical-shaped Ag NPs with diameters ranging from 5 to 100 nm are used in inks [283]. An inkjet ink made from spherical shaped colloidal Ag NPs with a diameter of 5–7 nm distributed at 10 weight percent in α-terpineol was sintered at 300 °C on a hot plate in order to generate conductive streaks of 80 µm, indicating a resistance of 3 µΩ.cm [284]. The first effective lead-free nano-silver paste was developed as a replacement for lead solder. It was proposed to replace the high-temperature, lead-rich solder used in electronics with this lead-free silver paste. The pastes were used to join copper bases and silicon diode chips at 350 °C in a nitrogen environment without the need for additional pressure [285]. Spherical-shaped Ag NPs with a diameter of approximately 8.5 nm were synthesized to formulate Ag NPs paste. The elastic properties of Cu-to-Cu joint samples made by sintering Ag NPs paste at a low temperature have been analyzed. It was noted that Ag NPs could offer a potential lead free alternative for assembling large scale (≥10 mm^2^) Cu chips in electronic devices that operate at high temperatures [286].

For the production of moveable digital screens through printing with ink jet printers, compatible inks are highly required, so that Ag NPs, due to their uniformity and small size, are widely used in electronic devices [14]. H_2_ O_2_ reduces onto the exterior of Ag NPs in an attempt to integrate the conductivity of inks provided strong applications in inkjet printing [287]. It has been reported that water-based G/Ag NPs composite inks are most effective in flexible printed electronics [288]. Ag NPs-based conductive inks used on fabrics [289]. Nanocellulose is used to formulate water-based inks with Ag NPs. These conductive inks serve applications in screen printing. Using screen printing, near field communication (NFC) printed antennas were manufactured and mounted on a paper-based substrate (NC-coated Klabin), resulting in functional NFC antennas [290]. To achieve eco-friendly printed electronics, stable water-based Ag NPs conductive inks are formulated by the chemical reduction method, which can be applied to ink-jet printing [291]. Polyethylene glycol (PEG) and ethylene glycol (EG) are used as reductants in the formulation of Ag NPs conductive ink with OP-10 as a dispersant. Ag NPs show high dispersion efficiency with a size of 40 nm and a resistivity of about 5.1 × 10^−3^ Ω·cm. These inks have uses in ink-jet printing and in flexible electrodes [292]. Printing silver inks is used to generate flexible biosensors that accelerate the identification of antibiotic-free milk without labels using inkjet printing [293].

#### 3.3.2. Transparent Conductive Films (TCFs)

In the past, indium tin oxide (ITO) was used as a transparent conductive film, but due to a lot of disadvantages, many alternatives were made, such as light-emitting devices, touch panels, solar cells, and displays. For the formation of high-performance TCF, perforated Ag NPs panels are a viable alternate material [277]. In terms of electrical conductivity, stiffness, and visibility, Ag NPs grid-based TCF operate extremely well, due to which they are widely used in optoelectronic devices [277].

For the production of transparent conductive films, Ag-r GO provides strong applications. Due to extraordinary stability, there are a large number of benefits in electronics [294]. Flexible transparent conductive film (FTCF) manufactured with Ag NWs ink provides strong potential in electrical conductors and provides high optical properties. They provide a variety of applications in transparent conductive films, such as in touch panels, solar cells, and many other applications [276]. Ag NWs film has 4000 times more conductivity as compared to Ag NPs film. Ag NPs film that is sintered at 300 °C has a higher resistivity than the film of long Ag NWs dried at 70 °C [295]. Scalable bar-coating method used to prepare flexible TCFs, possessing haze (1.04%) ITO TCFs with relatively small resistance of sheet (24.1 Ω/sq at 96.4% transmittance). Ag NWs are the most intriguing substance of all metal-based TCF alternatives with regards to haze because of the diameters ranging from 45 to 400 nm for other metal nanotroughs and metal grids. Haze has a direct relationship with the diameter of the structure [296]. The conductivity will increase with the increase in concentration of Ag NWs (printed in FCTF with Ag NWs ink); despite this, the light transmittance of FCTF decreases, which is related to its conductive process [276].

#### 3.3.3. Plasmonic Devices

For the development of biosensors with electrochemical properties, Ag NPs could be used as electrode substrates. For reducing the cost of biosensors, Ag NPs are most effective with high conductivity, LSPR, and sensibility as compared to the other metallic NPs [277]. The excellent and unique properties of Ag NPs provide exceptional applications in chemo-sensing and bio-sensing [297]. Due to their dynamic optical properties, Ag NPs ensure strong interactions of matter with light as they contain the majority of polarized electrons (plasmonic waveguides) [298]. Ag-based plasmonic NPs provide a lot of variety in biosensing [244]. Cubical-shaped plasmonic Ag NPs enhance the harvesting of light in PCS devices. Ag nanospheres show less enhancement in absorption of light as compared to cubical-shaped Ag NPs of the same size. Consequently, anisotropic Ag NPs provide high absorption enhancement with good performance in Personal Communications Service (PCS) devices [299,300,301]. Flower-like Ag NPs show high sensitivity to Rhodamine 6G while being used as SERS substrates. For the fabrication of LSPR biosensors, silver nanoplates (Ag NPIs) are considered intriguing shapes. The optical resonance of Ag NPIs is regulated around 500–1100 nm by modulating the thickness and diameter of the plate [244,302].

## 4. Challenges and Future Perspectives

As discussed previously, Ag NPs have been synthesized using various techniques. These techniques are categorized into two approaches: the top-down approach and the bottom-up approach. The top-down approach includes physical methods, while the bottom-up approach consists of biological and chemical methods. However, the biological synthesis approach is considered more suitable due to its lower toxicity and environmental friendliness, as it avoids the use of toxic chemicals. Microorganisms such as fungi and bacteria are also used for synthesizing Ag NPs, offering numerous advantages in a variety of applications such as biomedical, biosensing, electronics, textiles, and many other fields [13]. However, the biological approach involving microorganisms and plants requires specialized steps for cultivation and extraction. In short, it is a very labor-intensive procedure [184]. This process presents challenges in balancing traditional effectiveness against physical and chemical methods. Nevertheless, for achieving high production rates, biological techniques must be applied more extensively [118]. It is evident that the size and shape of Ag NPs strongly influence their properties. Ag NPs exhibit different properties based on their shape and size, with morphology entirely dependent on the synthetic techniques used. However, for applications such as biological imaging and solar energy processing, further modifications are still needed. There are still undefined factors that must be controlled to optimize the properties of Ag NPs [20]. As discussed earlier, the shape and size of Ag NPs significantly impact their applications. Literature reports various chemical approaches for the formation of Ag NPs, but only a few methods effectively control morphology, which is crucial for optimizing Ag NPs’ applications [6]. On the other hand, green techniques are beneficial, but sometimes they are unable to provide a strong grip on these two factors, which can prevent Ag NPs from exhibiting desired properties in the field of engineering [303]. It is still uncertain which parameters in biological synthetic approaches are responsible for the morphology of Ag NPs, although multiple studies have reported the production of different shapes of Ag NPs, such as spheres, flowers, triangles, and cubes [165].

### 4.1. Emerging Trends in Ag NPs Research

Ag NPs exhibit excellent properties that enable extraordinary applications in a variety of fields. However, further improvements are needed, such as avoiding the use of toxic chemicals, employing simple and cost-effective techniques, and ensuring high quality [304]. Research works on the synthesis of Ag NPs using viruses have been conducted. Viruses are composed of nucleoproteins that provide strong surface interactions with metals. One notable example is the green synthesis of Ag NPs using tobacco mosaic virus [305,306]. Ag NPs exhibit strong antimicrobial activities due to their large surface area. They are commonly used in everyday products such as pharmacy, food, cosmetics, fabrics, and various industries [13].

Rodríguez-Félix, F et al. demonstrated that Ag NPs hold significant potential for application in the food industry due to their ability to inhibit a wide range of pathogenic and spoilage bacteria. In their study, they employed a green synthesis method using aqueous extract from safflower (*Carthamus tinctorius* L.) waste, which not only allows for the production of Ag NPs with antimicrobial properties but also contributes to sustainability by reducing environmental pollution. The synthesized nanoparticles, which were uniform and spherical with an average diameter of 8.67 ± 4.7 nm, exhibited effective antibacterial activity against *Staphylococcus aureus* (Gram-positive) and *Pseudomonas fluorescens* (Gram-negative) even at low concentrations of 0.9 μg/mL, suggesting their potential application as antibacterial agents in the food and medical industries [307].

### 4.2. Impacts on Environment and Economy

In contrast to their various advantages in multiple fields, Ag NPs could be toxic. It has been reported that the toxicity of Ag NPs is also dependent on their shape and size [308]. Ag NPs pose a higher risk compared to larger elements due to the generation of reactive oxygen species (ROS). Smaller-sized Ag NPs (5 nm) have been reported to be more toxic than larger-sized Ag NPs (20–50 nm) [157]. Ag NPs are highly dangerous for marine insects, freshwater organisms, and fish due to ingestion and interactions with metals and ligands. They are also toxic to mammalian cells, affecting organs such as the lungs, brain, and skin [309]. Various studies have reported the production of Ag ions in biological approaches. It has been reported that Ag ions are responsible for the production of ROS. Research has also shown that Ag ions lead to the creation of superoxide radicals [310]. In addition, due to their surface adherence, Ag NPs can independently damage multiple cell functionalities [311]. The emission of Ag ions is dangerous for humanity as well as for nature. The use of organic materials can control the emission of Ag ions by absorbing them on Ag NPs. Furthermore, by focusing on the production mechanisms and modulating the properties of Ag NPs, hazardous effects could be reduced [312,313].

Ag NPs, due to their vast applications in fields like biomedical and engineering, face challenges related to environmental and economic issues. Ag NPs belong to noble metals, which are costly. Therefore, while they are easily produced on a small scale, generating them on a large scale is very difficult. The use of green methods could improve cost-effectiveness [314]. A recent study reported the growth-promoting properties of Ag NPs, which demonstrate potential benefits for both the economy and the environment [315].

Currently, investigations are focusing on the economic ambiguity associated with green synthetic techniques, which show high efficacy but lack sustainability on larger scales [316]. Agricultural food waste materials such as banana, pomegranate, orange, lemon, and tangerine peels have been successfully utilized for the synthesis of Ag NPs. This method is simple, rapid, inexpensive, and non-toxic. Plant extracts act as reducing, capping, and stabilizing agents, eliminating the need for external hazardous reducing agents [317]. An effective, cost-efficient, and sustainable alternative to conventional methods could be the environmentally friendly production of metal nanoparticles and their oxides from food waste. The green synthesis of Ag NPs from agricultural waste represents an advancement over chemical and physical methods. It is environmentally friendly, cost-effective, and can be easily scaled up for large-scale production of nanoparticles, all without the need for high temperatures, pressures, excessive energy, or toxic chemicals [318].

## 5. Conclusions

This review has provided a comprehensive overview of the different synthetic approaches to Ag NPs, including their pros and cons. Ag NPs can be synthesized by different methods, such as biological, chemical, and physical. Each method has unique benefits in terms of regulating the size, shape, and functionality of NPs. Biological methods using plant extracts, bacteria, and fungi are harmless to the environment, but the stability of Ag NPs is affected by factors such as organism characteristics, catalytic order, and inheritance. The most effective method is chemical reduction, which provides a high creation rate, thermostability, and stabilizer-tunable properties, but it can be erosive, energy intensive, and produce chemical contaminants. Physical methods such as sputtering, laser ablation, and arc discharge produce Ag NPs with high purity and uniformity in size without the use of hazardous chemicals, but they require expensive equipment and provide a low production rate. Whereas the photochemical method might be costly and time consuming, it provides homogeneity, uniform size, and minimal agglomeration using lasers, UV radiation, and sunlight.

Comprehending the structural characteristics is necessary because properties are intrinsically related to the characteristics. Depending on synthesis methods, Ag NPs have various shapes, such as spheres, cubes, wires, flowers, prisms, and pyramids. FCC is the most observed structure of Ag NPs. An important coverage of this review is the dependence of the properties of Ag NPs on size and shape, explicating their role in numerous applications. The SPR of Ag NPs depends upon the morphology, which affects the optical properties of NPs. Antimicrobial properties of Ag NPs are also size dependent; smaller Ag NPs show high antimicrobial properties due to their large surface area.

Ag NPs have excellent biological, chemical, and physical properties that make them beneficial in a variety of applications. Their unique optical properties, conductivity, and catalytic abilities have been made productive in fields ranging from electronics to environmental remediation. Additionally, their strong antimicrobial properties make them advantageous in a range of medical applications. Because of their high surface to volume ratio, Ag NPs have remarkable anticancer, antibacterial, and drug delivery activities. Ag NPs play a vital role in the enhancement of the electronics field, such as in conductive inks, sensors, and printed electronics. However, issues like uniformity, precisely controlled size, and environmental toxicity are pertinent and cause concern. It is essential to resolve the issues related to environmental toxicity and affordability. Future research must focus on enhancing the properties of Ag NPs with precise control over morphology, innovative development in synthetic techniques, and significant implementations in various fields. Furthermore, the review highlights the potential of Ag NPs to revolutionize several industries while underscoring the ongoing research that is essential to tackle challenges with large scale production, toxicity, and environmental impacts. The use of plants, bacteria, and agricultural/food waste, which together offer a comprehensive green synthesis strategy, is considered one of the most promising approaches. Utilizing food and agricultural waste for Ag NP synthesis is particularly interesting. This approach not only provides an ecological alternative to traditional chemical synthesis but also contributes to a circular economy by offering an effective waste management strategy. Agri-food waste contains several natural reducing agents, such as polyphenols, sugars, and proteins, making it an excellent choice for the synthesis of Ag NPs due to its intrinsic properties. This method allows for the large-scale production of Ag NPs with minimal environmental impact, paving the way for further advancements in nanotechnology and sustainable development. However, further research is needed to fully understand how biological constituents affect the properties of the generated NPs, scale up production, and improve biosynthetic methods.

## Figures and Tables

**Figure 1 nanomaterials-14-01425-f001:**
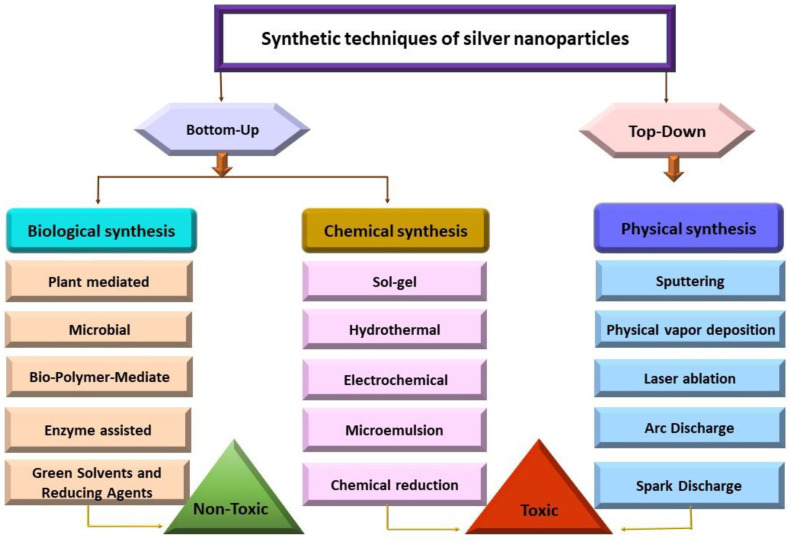
Schematic diagram for different synthetic techniques of Ag NPs.

**Figure 2 nanomaterials-14-01425-f002:**
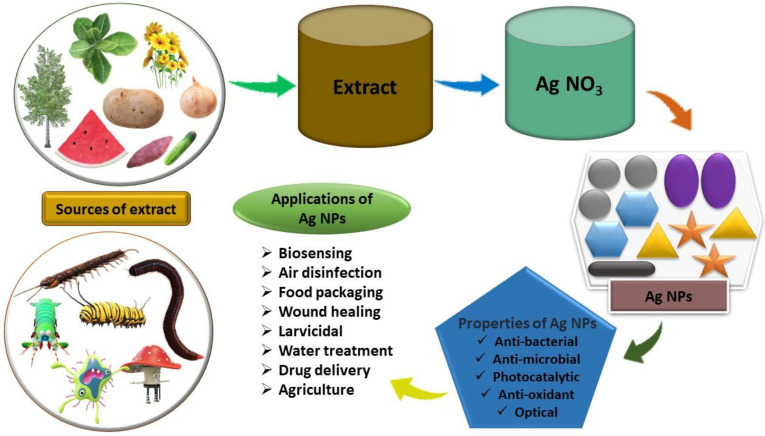
Schematic diagram for biological synthesis of Ag NPs.

**Figure 3 nanomaterials-14-01425-f003:**
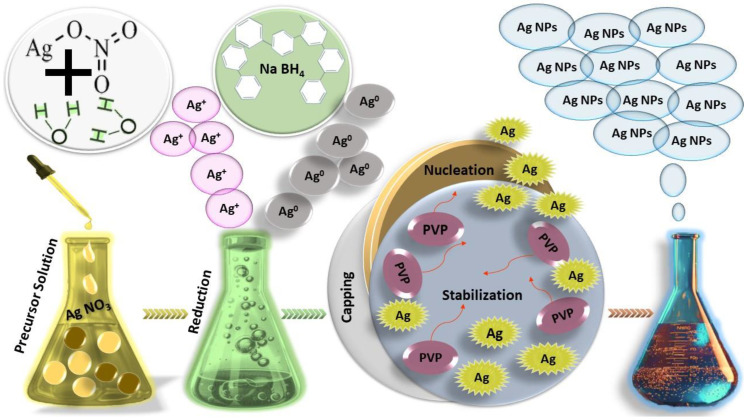
Schematic diagram for chemical synthesis of Ag NPs.

**Figure 4 nanomaterials-14-01425-f004:**
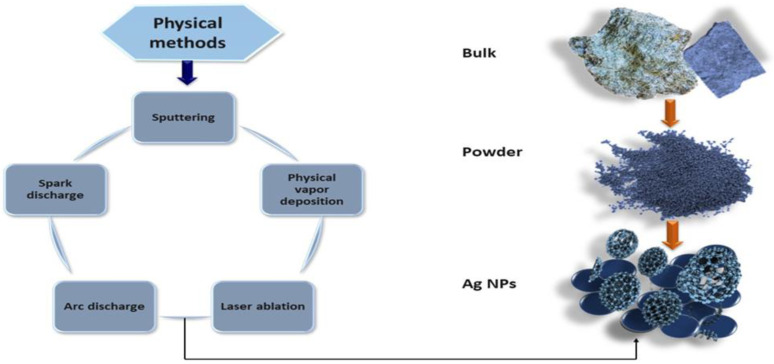
Schematic diagram for physical synthesis of Ag NPs.

**Figure 5 nanomaterials-14-01425-f005:**
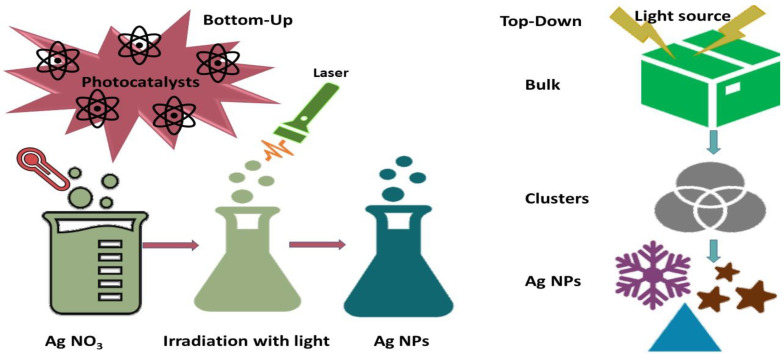
Photochemical synthesis of Ag NPs.

**Figure 6 nanomaterials-14-01425-f006:**
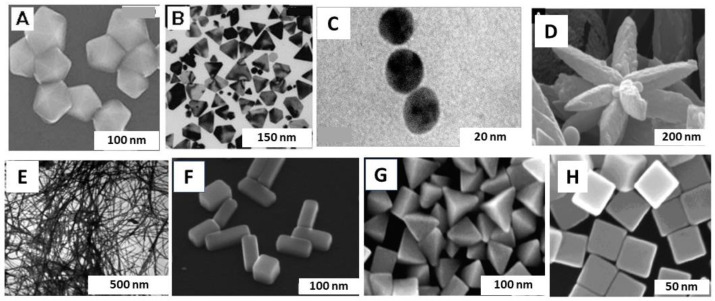
TEM and SEM images for different shapes of Ag NPs: (**A**) Decahedrons, (**B**) Prisms. Adopted from [204], (**C**) Sphere, (**D**) Flower, (**E**) Nanowires, (**F**) Nano-bars, (**G**) Pyramids, (**H**) Nano-cubes. Adopted from [19,205].

**Figure 7 nanomaterials-14-01425-f007:**
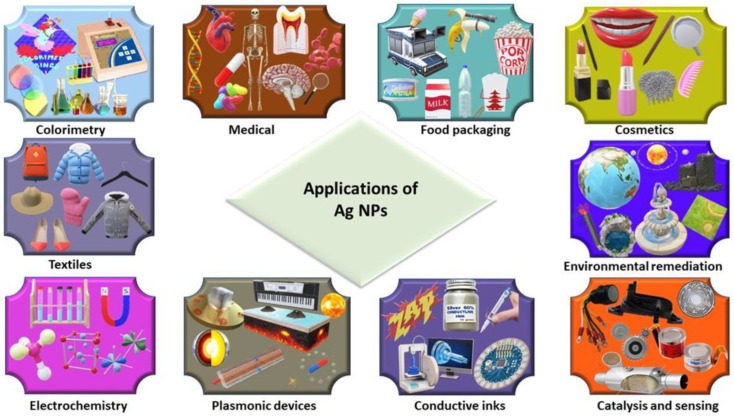
Schematic diagram for different applications of Ag NPs.

**Table 1 nanomaterials-14-01425-t001:** Pros and Cons of chemical methods.

Method	Advantages	Disadvantages	References
Chemical reduction	Versatility in structure Use of simple machinery Easy to handle No aggregation Good production rate Cost-effective method	Sintering of metal due to high heating Production of large size Ag NPs Use of hazardous compounds	[14,20,166]
Chemical vapor deposition (CVD)	Highly controllable technique High control on morphology, crystal structures, and on production rate Versatile in repetition throughout the synthetic process Homogeneous, rigid and exceptionally pure NPs	Usage of toxic, expensive and flammable chemicals Production cost increases with the use of various CVD variants Production rate eventually affected by temperature of substrate and pressure of gas	[167,168,169,170]
Microemulsion	High control on size Uniformity in morphology of Ag NPs Versatile in morphology selection	Use of large amounts of reagents. Costly method	[171,172,173]
Photochemical reduction	Simple to use Large scale production with small size NPs Less use of unsafe chemicals	Chances of impurities Lengthy and expensive process	[14,146,171]
Electrochemical	Large production rate of NPs Control on morphology by regulating the electrolysis variables Homogeneity improved by adjusting the concentration of solvents in electrolytes Cost effective	Production of impurities with NPs Un-stability in process Lengthy process Large consumption of energy Suitable for small level production	[14,18,71]
Hydrothermal	Cost-effective Versatile in structure Control on morphology Highly crystalline structures of nanocrystals	High energy consumption Small production rate Lengthy procedure Hard to handle Difficult to analysis crystal development directly Not flexible in reproducibility Space and energy consuming Lengthy process for thermal stability Furnace is needed to control heat which require more energy consumption and time	[4,14,18,174,175]
Microwave assisted	Easy method Less time required for whole process Large yielding rate	Costly technique Not suitable technique for NPs production	[176]
Polyol process	Ensures stability of NPs Generated Ag NPs are uniform	Slightly changes in synthetic substances affect the parameters of synthesized Ag NPs	[113]
Sol–gel method	Uniformity Homogeneous Control on size by adjusting the quantity of reactant and temperature	Time consuming Post treatments In drying, the products often compress and recess, hard to fabricate monolith product Aggregation is produced	[170,177,178]

**Table 2 nanomaterials-14-01425-t002:** Pros and Cons of physical methods.

Method	Advantages	Disadvantages	References
PVD (Evaporation/Condensation)	No use of solvents Low melting point materials Preferable in prolonged experiments	Use of tube furnace Heat Production Energy consumption Long time required for fabrication process	[26,171,177]
Sputtering	Control on morphology Low temperature Consistency in sputtered material High purity Less expensive than lithography	Impact of procedure on optical properties and morphology Heat production Low production rate	[4,124,165,179,180,181]
Laser ablation	Purity No use of reagents Environment friendly Precise control on size of NPs by adjusting laser parameters Reactivity and anti-microbial activities Ligand-free noble NPs produced by LA production using solvents in a wide range of solutions	Low production rate Effect of laser parameters on properties Large energy needed to get high ablation efficacy Highly dispersed lasers also unable to produce Ag NPs on industrial level Efficiency of ablation reduces due to scattering of NPs	[13,26,118,165,182]
Arc discharge	Quick and easy method Provides precise control on shape and size of Ag NPs	Structure, pureness and stability of created Ag NPs affected by the use of synthetic substances	[116]
Lithography	High control on morphology Good production rate Homogeneous Versatility in material	Laborious and complex technique Costly equipment required	[117,183]

**Table 3 nanomaterials-14-01425-t003:** Properties and structure of Ag NPs from different synthetic methods.

Synthetic Approaches	Sub-Methods	Size (nm)	Structure	Properties	References
Biological Synthesis	Plant-mediated synthesis	33.8	Spherical	Anti-bacterial/ Anti-oxidant	[27]
		25	Spherical	Anti-bacterial	[210]
		11–26	Spherical	Photocatalytic	[211]
		4–32	Spherical	Anti-oxidant/Larvicidal	[212]
		10–90	Spherical	Anti-bacterial	[213]
		42.71 ± 17.97	Spherical	Anti-Cancer	[214]
		6–45	Spherical	Anti-bacterial	[215]
		˂100	Cubic	Anti-bacterial	[216]
		14–24	Spheroid	Anti-oxidant	[217]
		26–39	Spherical	Anti-microbial/Anti-oxidant/photocatalytic	[218]
	Microbial synthesis	20–50	Spherical	Optical	[31]
		40–50	Spherical	Anti-oxidant/ Antibacterial	[32]
		10–60	Spherical/cubic	Anti-proliferative	[217]
		10–30	Spherical	Anti-bacterial	[219]
		10–40	Irregular	Anti-bacterial	[220]
		8–90	Spherical	Anti-microbial	[221]
		14.0 ± 4.7	Spherical	Antibacterial	[38]
	Bio-Polymer Mediated	10–50	Spherical	Anti-bacterial	[39]
	Enzyme-assisted Synthesis	10–20 (TEM)\5–10 (XRD)	Spherical	Anti-bacterial	[42]
		10–50	Spherical	Anti-bacterial	[45]
Chemical synthesis	Bromide-mediated Polyol process	20	Nanowires (penta-twinned)	Conductive	[52]
	Sol-gel	7–8	_	Catalytic	[57]
		15–20	_	Anti-oxidant	[222]
		20	_	Optical/ Plasmonic	[59]
	Hydro-thermal method	17.1 ± 5.9	_	Anti-bacterial	[60]
		5	Spherical	Catalytic	[223]
		29	Spherical	Anti-fungal	[61]
		70.70 ± 22–192.02 ± 53	Spherical	Anti-bacterial	[224]
		23–48	Spherical	Anti-bacterial	[225]
		7.1	Quasi-spherical	Anti-viral	[226]
		3–10	Spherical	Catalytic	[74]
	Chemical Reduction	68	_	Anti-Microbial	[83]
		35–80	Quasi-spherical	Electrical Conductivity	[85]
		10–30	Spherical	Not reported	[227]
		10–250	Spherical		[228]
		6.18 ± 5	_	Anti-Microbial	[86]
		10–100	Spherical	Optical/Catalytic/Anti-microbial	[229,230]
		50–200 (edge-length)	Pyramids	Plasmonic	[19]
	Polyol Process	50–100	Hexagon	Anti-Microbial	[97]
		80–150	Icosahedral	Optical	[104]
		420–430	Spherical	Anti-bacterial	[103]
		35–45	Oval like Spherical	Photo-catalytic/Anti-bacterial/ Anti-fungal	[106]
		79–200	Spherical	Catalytic	[107]
Physical methods	Sputtering	˂10	Wormlike	Catalytic	[122]
	Laser ablation	20–50	Spherical	Anti-microbial	[135]
		17	Spherical	Physicochemical	[231]
		7.5–12	Spherical	Optical	[232]
		25–40	Spherical	Optical	[233]
	Arc discharge	17	Spherical	Anti-bacterial	[234]
		72	Spherical	Optical	[235]
		19	Cubic	Anti-microbial	[236]
		20–30	Spherical	_	[237]
Photo-chemical synthesis		40–220	Prism/decahedron/ Plate	_	[238]
		31.4 ± 1.4	Triangular plate	Optical	[239]
		Aprox.8.6	Spherical		[240]
		0.74–1.12	Spherical	_	[241]
		4–20	Rods/polyhedrons/ Spheres	_	[242]

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
