# Peer review of "Silver Nanoparticles: Synthesis, Structure, Properties and Applications"

_nanomaterials, 2024, doi:10.3390/nano14171425_

Round 1

Reviewer 1 Report

Comments and Suggestions for Authors

In this paper, the authors reviewed the synthesis methods of silver nanoparticles, their properties, and applications. The topic is interesting; however, the paper requires a major revision before being reconsidered for publication. The following points should be addressed:

1.The scope of the paper is identical to several recently published review articles, e.g., https://doi.org/10.1016/j.microc.2023.109615. Therefore, I do not see a point in writing a review on this topic again. The authors must clearly explain what makes their article unique compared to previously published review articles. Furthermore, they should explain why another review of Ag NPs is needed and what is the novelty.

2.If the novelty is the review of environmentally friendly synthesis methods, as suggested in Fig. 2, it must be clearly highlighted.

3.The authors should discuss the size distribution and particle size uniformity when describing the various synthesis methods.

4.The authors discuss the use of Ag NPs in electronics (p. 20); however, they exclude solders. The application of Ag NPs as lead-free solder materials should be mentioned in the manuscript. See, e.g., http://dx.doi.org/10.1007/s13391-013-3148-5.

Technical issues:

5.The authors should use italics when writing the Latin names of plants and other living species (top of page 6).

6.It is necessary to number the lines of the manuscript. It helps reviewers to point to specific portions of the manuscript where a revision is necessary.

7.Several abbreviations are unexplained at their first mention in the manuscript, e.g., PLEC, SERS, MIE theory, etc. All abbreviations should be defined.

8.Fig. 6 can be removed. Identical information is presented in Fig. 7 and Table 3.

Author Response

Point by Point Response to Reviewers Comments

Reviewer #1:

Comment 1: The scope of the paper is identical to several recently published review articles, e.g., https://doi.org/10.1016/j.microc.2023.109615. Therefore, I do not see a point in writing a review on this topic again. The authors must clearly explain what makes their article unique compared to previously published review articles. Furthermore, they should explain why another review of Ag NPs is needed and what is the novelty.

Response 1: Thank you for raising this point. We are aware that many review articles on silver nanoparticles (Ag NPs) are available in the literature, covering various parameters. In our review article, we aimed to provide a comprehensive overview that addresses all aspects related to Ag NPs, including synthesis methods (with their advantages and disadvantages), the dependence of properties on synthetic methods and morphology, and their applications. This will be helpful for researchers to obtain the necessary information and easily select the appropriate synthetic method based on their required properties and applications. Additionally, our review article introduces novel insights into the use of biological methods, including the utilization of agricultural and food waste, which could contribute to the development of new environmentally friendly techniques for Ag NPs synthesis. This approach would be beneficial for the future applications of Ag NPs in various fields.

Comment 2: If the novelty is the review of environmentally friendly synthesis methods, as suggested in Fig. 2, it must be clearly highlighted.

Response 2: Thank you for highlighting this point. In this review, we aim to showcase the novelty of environmentally friendly synthesis techniques using natural resources or waste from the agricultural and food industries. Further explanation is provided in this regard under the following sections: Subtopic 2.1, 'Biological methods' (page 4, lines 158-167); Section 4.1, 'Emerging trends in Ag NPs research' (page 24, lines 834-843); and the Conclusion (lines 917-929).

Comment 3: The authors should discuss the size distribution and particle size uniformity when describing the various synthesis methods.

Response 3: We agree with your comment that the main focus in most research articles is on applications. We have added information about size distribution and particle size uniformity from the research articles where this was explained. We understand that it may be challenging to locate the specific lines and pages for each modification in synthetic methods. Therefore, we have highlighted the revisions made in response to this comment in blue.

Comment 4: The authors discuss the use of Ag NPs in electronics (p. 20); however, they exclude solders. The application of Ag NPs as lead-free solder materials should be mentioned in the manuscript. See, e.g., http://dx.doi.org/10.1007/s13391-013-3148-5.

Response 4: We are thankful to you for highlighting this point. We missed an important piece of information. The revision made in Subtopic 3.3.1, 'Conductive inks' (page 22, lines 728-737).

Comment 5: The authors should use italics when writing the Latin names of plants and other living species (top of page 6).

Response 5: We changed the Latin names of plants and other living species to italics.

Comment 6: It is necessary to number the lines of the manuscript. It helps reviewers to point to specific portions of the manuscript where a revision is necessary.

Response 6: We agree with this point. Line numbers have been added to every line in the manuscript.

Comment 7: Several abbreviations are unexplained at their first mention in the manuscript, e.g., PLEC, SERS, MIE theory, etc. All abbreviations should be defined.

Response 7: The modification has been completed.

Comment 8: Fig. 6 can be removed. Identical information is presented in Fig. 7 and Table 3.

Response 8: We have removed Figure 6 from the manuscript.

Reviewer 2 Report

Comments and Suggestions for Authors

Dear MDPI Editor, thank you for allowing me to be part of the Nanomaterials MDPI review group and to improve the quality of the manuscript entitled " Silver nanoparticles: synthesis, structure, properties and applications". I have reviewed the manuscript and consider it suitable for publication after the authors make the following adjustments and respond to the comments.

Comment to authors

1.       In general, throughout the manuscript, include which waste and by-products from the agricultural sector and/or food industry are currently recognized within the biological methods for the synthesis of silver nanoparticles.

2.       Within subtopic 2.1 biological methods, include a paragraph that talks about green synthesis from agro-industrial waste as waste and by-product from both the agricultural sector and the food industry.

3.       Include the following information: The green synthesis of silver nanoparticles involves using agricultural waste and by-products from the food industry as eco-friendly reducing and stabilizing agents. Examples of such waste include banana peels, orange peels, and potato skins, which are rich in natural compounds like polyphenols and flavonoids. These compounds reduce silver ions into nanoparticles and help stabilize them, minimizing the need for harmful chemicals. This approach not only provides a sustainable method for nanoparticle synthesis but also adds value to agricultural and industrial waste, contributing to waste management and environmental protection. cite to: (2022). Trends in Sustainable Green Synthesis of Silver Nanoparticles Using Agri‐Food Waste Extracts and Their Applications in Health. Journal of Nanomaterials, 2022(1), 8874003.

4.       Include the following paragraph as a discussion of the synthesis of agri-food waste: López-Cota et al. (2021) demonstrated that silver nanoparticles (AgNPs) hold significant potential for application in the food industry due to their ability to inhibit a wide range of pathogenic and spoilage bacteria. In their study, they employed a green synthesis method using aqueous extract from safflower (Carthamus tinctorius L.) waste, which not only allows for the production of silver nanoparticles with antimicrobial properties but also contributes to sustainability by reducing environmental pollution. The synthesized nanoparticles, which were uniform and spherical with an average diameter of 8.67 ± 4.7 nm, exhibited effective antibacterial activity against Staphylococcus aureus (Gram-positive) and Pseudomonas fluorescens (Gram-negative) even at low concentrations of 0.9 μg/mL, suggesting their potential application as antibacterial agents in the food and medical industries. cite: (2021). Recovery of phytochemicals from three safflower (Carthamus tinctorius L.) by‐products: Antioxidant properties, protective effect of human erythrocytes and profile by UPLC‐DAD‐MS. Journal of Food Processing and Preservation, 45(9), e15765.(2021); Sustainable-green synthesis of silver nanoparticles using safflower (Carthamus tinctorius L.) waste extract and its antibacterial activity. Heliyon, 7(4).

5.       Include more references on the use of waste and by-products for the green synthesis of silver nanoparticles

Author Response

Point by Point Response to Reviewers Comments

Reviewer #2:

Comment 1: In general, throughout the manuscript, include which waste and by-products from the agricultural sector and/or food industry are currently recognized within the biological methods for the synthesis of silver nanoparticles.

Response 1: Thank you for mentioning this point. The data and references related to waste and by-products from the agricultural sector and/or food industry are included throughout the manuscript as follows: Subtopic 2.1 Biological methods; (References 24 and 25), page 4, lines 158-167; Subtopic 2.1.1, 'Plant-mediated synthesis' (References 33-35), page 6, lines 208-220; Subtopic 2.5, 'Pros and Cons of Different Synthetic Approaches to Ag NPs' (Reference 25), page 13, lines 505-508; Subtopic 2.6, 'Structure and Properties of Ag NPs' (References 202 and 203), page 17, lines 584-590; Table 3, 'Properties and Structure of Ag NPs from Different Synthetic Methods' (References 214-216), page 18; Subtopic 3.1.1, 'Antiseptics' (Reference 254), page 20, lines 644-650; Subtopic 3.1.3, 'Imaging and Diagnostics' (Reference 265), page 21, lines 675-678; and Subtopic 4.2, 'Impacts on Environment and Economy' (References 322 and 323), page 25, lines 868-878.

Comment 2: Within subtopic 2.1 biological methods, include a paragraph that talks about green synthesis from agro-industrial waste as waste and by-product from both the agricultural sector and the food industry.

Response 2: In subtopic 2.1 biological methods, the information provided by reviewer added with one more reference; page 4, lines 158-167.

Comment 3: Include the following information: The green synthesis of silver nanoparticles involves using agricultural waste and by-products from the food industry as eco-friendly reducing and stabilizing agents. Examples of such waste include banana peels, orange peels, and potato skins, which are rich in natural compounds like polyphenols and flavonoids. These compounds reduce silver ions into nanoparticles and help stabilize them, minimizing the need for harmful chemicals. This approach not only provides a sustainable method for nanoparticle synthesis but also adds value to agricultural and industrial waste, contributing to waste management and environmental protection. cite to: (2022). Trends in Sustainable Green Synthesis of Silver Nanoparticles Using Agri‐Food Waste Extracts and Their Applications in Health. Journal of Nanomaterials, 2022(1), 887400

Response 3: We are thankful for providing this useful information; The given information has been added in Subtopic 2.1, 'Biological Methods,' on page 4, lines 160-167

Comment 4: Include the following paragraph as a discussion of the synthesis of agri-food waste: López-Cota et al. (2021) demonstrated that silver nanoparticles (Ag NPs) hold significant potential for application in the food industry due to their ability to inhibit a wide range of pathogenic and spoilage bacteria. In their study, they employed a green synthesis method using aqueous extract from safflower (Carthamus tinctorius L.) waste, which not only allows for the production of silver nanoparticles with antimicrobial properties but also contributes to sustainability by reducing environmental pollution. The synthesized nanoparticles, which were uniform and spherical with an average diameter of 8.67 ± 4.7 nm, exhibited effective antibacterial activity against Staphylococcus aureus (Gram-positive) and Pseudomonas fluorescens (Gram-negative) even at low concentrations of 0.9 μg/mL, suggesting their potential application as antibacterial agents in the food and medical industries. cite: (2021). Recovery of phytochemicals from three safflower (Carthamus tinctorius L.) by‐products: Antioxidant properties, protective effect of human erythrocytes and profile by UPLC‐DAD‐MS. Journal of Food Processing and Preservation, 45(9), e15765.(2021); Sustainable-green synthesis of silver nanoparticles using safflower (Carthamus tinctorius L.) waste extract and its antibacterial activity. Heliyon, 7(4).

Response 4: We are very grateful to you for this paragraph. The information is very interesting and will enhance our review article. The given paragraph has been added to Subtopic 4.1, 'Emerging Trends in Silver Nanoparticles Research,' on page 25, in the last paragraph, lines 834-843.

Comment 5: Include more references on the use of waste and by-products for the green synthesis of silver nanoparticles.

Response 5: We have added more references on the use of waste and by-products for the green synthesis of silver nanoparticles, as mentioned in response to Comment 1.

Note: Reviewer #1, revisions are highlighted in blue, and for Reviewer #2, revisions are highlighted in yellow in the manuscript.

Round 2

Reviewer 1 Report

Comments and Suggestions for Authors

Authors answered my comments and improved their manuscript. It can be accepted for publication.

Reviewer 2 Report

Comments and Suggestions for Authors

Accepted in current form